# Intracellular HINT1-Assisted Hydrolysis of Nucleoside 5′-*O*-Selenophosphate Leads to the Release of Hydrogen Selenide That Exhibits Toxic Effects in Human Cervical Cancer Cells

**DOI:** 10.3390/ijms23020607

**Published:** 2022-01-06

**Authors:** Agnieszka Krakowiak, Liliana Czernek, Marta Pichlak, Renata Kaczmarek

**Affiliations:** Department of Bioorganic Chemistry, Centre of Molecular and Macromolecular Studies Polish Academy of Sciences, Sienkiewicza 112, 90-363 Lodz, Poland; lczernek@cbmm.lodz.pl (L.C.); marta.pichlak007@gmail.com (M.P.); renata@cbmm.lodz.pl (R.K.)

**Keywords:** HINT1-HIstidine Triad nucleotide-binding protein 1, (d)NMPSe-nucleoside selenophosphate, (d)NMPS-nucleoside thiophosphate, cancer, HIT

## Abstract

In this study, we present a new selenium derivative, 2′-deoxyguanosine-5′-*O*-selenophosphate (dGMPSe), synthesized by the oxathiaphospholane method and adapted here for the synthesis of nucleoside selenophosphates. Using biochemical assays (HPLC- and fluorescence-based), we investigated the enzymatic activity of HINT1 towards dGMPSe in comparison with the corresponding thiophosphate nucleoside, i.e., dGMPS. Both substrates showed similar k_cat_ and a small difference in K_m_, and during the reactions the release of reducing agents such as H_2_Se and H_2_S were expected and detected. MTT viability assay and microscopic analysis showed that dGMPSe was toxic to HeLa cancer cells, and this cytotoxicity was due to the release of H_2_Se. The release of H_2_Se or H_2_S in the living cells after administration of dGMPSe and/or dGMPS, both without carrier and by electroporation, was observed using a fluorescence assay, as previously for NMPS. In conclusion, our comparative experiments with dGMPSe and dGMPS indicate that the HINT1 enzyme is capable of converting (d)NMPSe to (d)NMP and H_2_Se, both in vitro and intracellularly. Since the anticancer activity of various selenium compounds depends on the formation of hydrogen selenide, the actual inducer of cell death, we propose that selenium-containing nucleotides represent another option as novel compounds with anticancer therapeutic potential.

## 1. Introduction

Selenium (Se) is an essential trace element that plays a crucial role in many cellular physiological processes that depend on the amount of selenium compounds intake. At physiologically achievable nanomolar concentrations, natural selenium-containing compounds are not toxic to mammalian cells, and are primarily used for the synthesis of selenoacid-containing proteins, i.e., selenocysteine (SeCys), the 21st member of the natural amino acid family and selenomethonine (SeMet). In higher concentrations, as observed with the intake of >400 µg Se/day, selenium can be lethal to cells [1]. As shown, both excessive and insufficient Se levels are associated with various human disorders (cancer, cardiovascular and neurodegenerative diseases) [2,3]. Interestingly, several selenium compounds in the low micromolar range have shown significant cytotoxicity [1], especially against various malignant cells (lung, prostate, cervical, colon and others), but the mechanism of their anticancer effect is not yet fully understood. Dietary selenium compounds, i.e., inorganic and organic selenium species, e.g., selenite, SeMet, SeCys, etc., are metabolized differently in vivo, and produce different selenium metabolites [2,4]. However, their metabolic pathways intersect at a common metabolite, which has been widely identified as hydrogen selenide (H_2_Se, mostly expressed as HSe^−^ at physiological pH), a reduced form of selenium [4]. Therefore, the biological function of selenium compounds and their toxicity/activity profiles are likely to be exerted mainly via their metabolites, with the observed effects depending on the dose and the specific chemical form (e.g., inorganic or organic), as well as the redox state and experimental model [2].

In 2002, sodium selenite (Na_2_SeO_3_; dietary selenium compound supplied with drinking water), which has been shown to produce H_2_Se during metabolism via glutathione reductase (and other cellular reduction systems), was shown to have antitumor effects through the generation of reactive oxygen species (ROS) [5]. H_2_Se is a highly reductive molecule with powerful chemical reactivity and short lifetime, and cannot be easily detected in cells or animal models. H_2_Se has been shown to react rapidly with O_2_ to form elemental selenium and superoxide anion radicals (O_2_^▪−^), which lead to DNA strand breaks and apoptosis of cancer cells [6,7]. On the other hand, a novel anticancer mechanism of pharmacological doses of Na_2_SeO_3_ as a source of H_2_Se after treatment of liver cancer HepG2 cells was discovered in 2019 [8]. It was shown that the accumulation of H_2_Se under hypoxic conditions resulted in reductive stress followed by autophagy-associated cell death. Thus, it can be concluded that H_2_Se is an essential molecule for the anticancer activity of selenium compounds.

Selenium and sulfur elements belong to the same family of chalcogens. S and Se atoms have similar atomic radii (88 and 103 pm), covalent radii (102 and 116 pm), and electronegativity (2.58 and 2.55, respectively). There are also some differences, as a sulfur atom generally forms much stronger covalent bonds, while a selenium atom is more easily oxidized. H_2_Se and H_2_S share similar physical and chemical properties, but also show important differences such that they exert different biological activity. The pK_a1_ value of H_2_S (6.88) and H_2_Se (3.89) differs significantly, while the pK_a2_ values (HX^−^) do not vary so much (14.15 and 15.1) [9]. Moreover, the half-life of H_2_S in air-saturated water (in the absence of other reactants) is a few minutes, whereas the half-life of H_2_Se at pH 7 is about 30 s.

Our previous study on the biological activity of 5′-*O*-thiophosphate nucleosides showed that these compounds can be a source of H_2_S both under in vitro and intracellular conditions [10,11,12]. These results led us to consider whether selenium congeners of thio-nucleotides can release H_2_Se in living cells and whether HINT1 (HINT1 means protein from *human*) is the enzyme responsible for this process, as shown previously [12]. This enzyme belongs to the Histidine Triad superfamily (HIT proteins) [13], was found in the cytoplasm of all organisms and acts as a nucleotide hydrolase. The enzymatic activity of Hint1 was suggested to be related to the hydrolysis of Lys-AMP, an activated lysine intermediate in tRNA aminoacylation, and the control of Ap4A signaling molecule levels [14]. In vitro, Hint1 (Hint1 has a broader meaning and means protein from species other than human, e.g., from *rabbit*) hydrolyzes mainly purine nucleoside substrates such as phosphoramidates (P-N [15]), mixed anhydrides (P-O-C(O)-R [16]), fluorophosphates (P-F [17]) and thiophosphates (P-S [10]).

In the present study, we report the chemical synthesis of a novel selenium derivative, 2′-deoxyguanosine-5′-*O*-selenophosphate (dGMPSe), and demonstrate through a series of in vitro Hint1-catalyzed enzymatic assays and an MTT cytotoxicity assay that this compound and H_2_Se, its hydrolytic product, induce death of human cervical cancer cells. Our data suggest that the tested compound and other nucleoside 5′-*O*-selenophosphates may represent a new option for cancer treatment.

## 2. Results

### 2.1. Chemistry

In this study, we present 2′-deoxyguanosine-5′-*O*-selenophosphate (dGMPSe, **4**), a new selenium derivative of 2′-deoxyguanosine monophosphate (dGMP), being synthesized by the oxathiaphospholane method (Figure 1) developed by Stec et al. for the stereocontrolled synthesis of oligonucleotide phosphorothioates [18] and phosphoroselenoate [19]. The 3′-*O*-acyl-*N2*-iBu-protected 2′-deoxyguanosine **1** was reacted with 2-chloro-4,4-pentamethylene-1,3,2-oxathiaphospholane **2** in the presence of *N,N*-diisopropylethylamine (DIPEA) and the resulting P(III) intermediate was oxidized with elemental selenium. Crude **3** was isolated from the reaction mixture by silica gel column chromatography and deprotected with anhydrous 3-hydroxypropionitrile and DBU. The final product **4**, purified by Sephadex A-25 ion exchange chromatography, was obtained in a 30% yield. Its structure was confirmed by ^1^H NMR and HRMS (ESI-TOF) mass spectrometry (Figure 1). The negative MS-ESI spectrum of *m*/*z* 409.9774 amu (A) confirms of dGMPSe calculated monoisotopic molecular mass MW = 409.9778. The spectrum shows the typical isotopic distribution for the selenium derivatives, as it contains a series of signals characteristic of products with selenium atoms in natural abundance: ^74^Se—0.86%, ^76^Se—9.26%, ^77^Se—7.60%, ^78^Se—23.69%, ^80^Se—49.80% and unstable ^82^Se—8.82% (Figure 1b).

### 2.2. Enzymatic Assays

#### 2.2.1. RP-HPLC Analysis of Reaction Products and Determination of dGMPSe Hydrolysis Rate Catalyzed by HINT1

HINT1-assisted hydrolysis of nucleoside 5′-*O*-thiophosphates ((d)NMPS) leading to the formation of nucleoside monophosphate ((d)NMP) and the release of H_2_S has been reported previously [10,20]. Based on the similar chemical properties of sulfur and selenium, we hypothesized that nucleoside 5′-*O*-selenophosphates ((d)NMPSe) are also the substrates for HINT1, and accordingly convert to nucleoside monophosphate and hydrogen selenide. To test our hypothesis, we performed several enzymatic experiments. Although a lower activity of 2′-deoxyribonucleotide than of ribonucleotide derivatives toward Hint1 was found previously, we chose 2′-deoxyriboguanosine-5′-*O*-selenophosphate (dGMPSe, **4**) here due to its relatively easy synthetic availability. Therefore, after incubation of 2′-deoxyguanosine 5′-*O*-selenophosphate with purified recombinant HINT1 hydrolase, the formation of the expected products, dGMP and H_2_Se, should be observed. Two different assays were used to detect these products. The first assay was based on HPLC and the second on fluorescence. In the first assay, the relative contents of the dGMPSe substrate and the corresponding dGMP product were analyzed by reversed-phase high-performance liquid chromatography (RP-HPLC) with the UV light product detection at λ = 254 nm. In this assay, we quantified dGMPSe and dGMP (Figure 2) and found the difference in their retention times (Rt) of about 0.8 min. dGMPSe in buffer alone appeared stable under the reaction conditions.

Moreover, the hydrolysis rate of **4** was compared with that of dGMPS and GMPS (Table 1). Similar hydrolysis rates (about 13 pmol × min^−1^ × µg^−1^) were observed for both 2′-deoxy series substrates (dGMPSe and dGMPS), while GMPS was a significantly better substrate (by one order of magnitude).

This assay was extended by analyzing the hydrolysis rate in the presence of HINT2, a homolog of HINT1 and member of the Hint protein family [13]. The enzyme showed 10-fold lower activity (approximately 1.55 pmol × min^−1^ × µg^−1^) towards both deoxyguanosine derivatives than HINT1. The activity towards GMPS was similar for both enzymes (ca. 270 pmol × min^−1^ × µg^−1^), which is consistent with previous reports on the affinity and activity of HINT2 towards dGMP, GMP, dGMPS and GMPS [21].

The kinetic parameters (k_cat_ and K_m_) were determined for the HINT1-catalyzed deselenation of dGMPSe and compared with those determined for the desulfuration of dGMPS and AMPS [10] (Figure 3, Table 2). The k_cat_/K_m_ ratio (47.6 M^−1^ s^−1^) for dGMPSe was slightly lower than for the two sulfur-containing substrates (65.3 and 81.0 M^−1^ s^−1^, respectively), and this small difference originates from K_m_, since all three k_cat_ values are around 0.02 s^−1^. According to the double displacement mechanism of hydrolysis catalyzed by Hint1 and its kinetic mechanism [20,22], the covalent GMP-enzyme complex **E*** is the identical intermediate for both substrates (dGMPSe and dGMPS; Figure 2), so the k_cat_ values determining the final steps of the hydrolysis reaction are similar and consistent with the Michaelis-Menten kinetics. In agreement with Hint1 preferences (higher activity towards ribonucleoside than towards 2′-deoxyribonucleoside phosphorothioates [10]), both deoxyguanosine derivatives showed a higher K_m_ value than AMPS. Moreover, dGMPSe showed a slightly higher K_m_ value than dGMPS, suggesting that the larger Se atom influences the initial steps of the enzymatic reaction, i.e., the association of the substrate with the enzyme in the active site and the formation **ES1a** or **ES1b**, and/or the formation of the covalent intermediate **E*** (Figure 2), in accordance with the kinetic mechanism of HINT1 reported in detail previously [22].

#### 2.2.2. Fluorescence Assay for the Analysis of HINT1-Catalyzed dGMPSe Hydrolysis Products

The release of the second product, hydrogen selenide, was detected by a fluorescence assay with fluorogenic probes SF7 and/or SF4 (ex. 485 nm, em. 528 nm) [23]. In the presence of reducing agents such as H_2_S or H_2_Se, these fluorogenic compounds become fluorescent. The usefulness of SF7 and SF4 for the detection of H_2_S in the hydrolysis reaction of nucleoside 5′-*O*-phosphorothioate has been previously demonstrated [12,23]. In this study, we found that after the addition of HINT1 to the reaction mixture containing dGMPSe and SF7/SF4, the fluorescence increased with time compared to the negative controls (Figure 4). Reactions without substrate or with dGMP were used as negative controls and mixtures containing GMPS or dGMPS were used as positive controls. dGMPSe in buffer alone appeared stable under the reaction conditions. Due to the higher chemical reactivity of H_2_Se than H_2_S (reductive properties) relative to SF7/SF4 probes, higher fluorescence values were observed for dGMPSe than for GMPS (Figure 4a) and dGMPS (Figure 4b) in these experiments. SF7 was more sensitive to H_2_Se than SF4: both probes were used at concentrations of 5 µM (fluorescence intensity 4000 versus 150, Figure 4a,b). Therefore, in further cellular assays, SF7 was used at a concentration of 2.5 µM and SF4 at a concentration of 5 µM.

Importantly, the HPLC assay showed higher activity of HINT1 towards ribonucleotide series substrates ([10], Table 2) and similar activity towards dGMPSe and dGMPS. Given the higher activity of H_2_Se than H_2_S towards both SF probes, the HPLC assay should be considered informative to address the activity of HINT1 towards different substrates.

We also used the same assay to compare the activity of HINT1 and HINT2 toward dGMPSe. To detect similar fluorescence intensity, it was necessary to use HINT2 at twice the molar ratio of HINT1. The data obtained (Figure 5) show that despite the higher molar ratio, HINT2 was significantly less active towards the used substrate than HINT1, which is consistent with previously reported data for this enzyme and dGMPS [21].

Therefore, it can be concluded that the hydrolysis of dGMPSe catalyzed by HINT1 (and HINT2) occurs under in vitro conditions, and the products of this reaction are 2′-deoxyguanosine phosphate and hydrogen selenide, according to the following scheme:



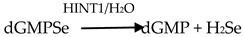



These results help us to show that both fluorogenic probes are susceptible to react with H_2_Se, and can be used for further cellular experiments.

### 2.3. Cellular Experiments

#### 2.3.1. Cytotoxic Effects of dGMPSe on HeLa Cells

As mentioned earlier, H_2_Se has been shown to have a chemotherapeutic effect on cancer cells. Therefore, our next goal was to investigate whether the nucleoside 5′-*O*-selenophosphates are toxic to cancer cells and to demonstrate that this effect is due to the release of H_2_Se (partially dissociated to HSe^−^ at physiological pH). To evaluate the cytotoxicity of dGMPSe, the MTT assay was performed on cervical cancer cells (HeLa), assessing cell viability at 12, 24 and 48 h. In this assay, a yellow tetrazole salt [3-(4,5-dimethylthiazol-2-yl)-2,5-diphenyltetrazolium bromide, MTT] is reduced by mitochondrial reductase to purple formazan, which can be quantified spectrophotometrically [24]. The absorbance of formazan is relative to the number of living cells. Any reducing agents present in the medium with the cells can interfere with the assay and affect the results. Since we anticipated the intracellular release of H_2_Se, the strong reducing agent that is suggested to be transported effectively via ATPases [25], we modified this method slightly. After the indicated time (12, 24, or 48 h) and just before the addition of the MTT, the medium was replaced with fresh and the assay was performed as usual. The viability of HeLa cells was determined at five different concentrations of dGMPSe: 0.1 µM, 1 µM, 10 µM, 50 µM and 100 µM. Cells treated with H_2_O were used as control (100% viability in MTT assay). The dGMP was used as an additional negative control. The dose-response curves were used to calculate the IC_50_ values (the concentration of compound that reduces cell growth by 50%), which are shown in Table 3. An example of the plot of viability at 12 and 24 h is shown in Figure 6. The results show that dGMPSe was toxic to HeLa cells after only 12 h, as the IC_50_ = 19 µM, whereas the IC_50_ after 24 h was 8 µM. This value is comparable to the cytotoxicity of Na_2_SeO_3_, the first dietary selenium compound shown to produce H_2_Se, as the inhibitory concentration was 5 μM after 24 h [8].

To check whether the effect of dGMPSe cytotoxicity observed by the MTT assay can be confirmed by another method, we used inverted microscopy of HeLa cells treated with dGMPSe (light microscope-phase contrast) and the fluorogenic SF7 probe (fluorescence microscope). Indeed, morphological changes were observed for HeLa cells already after 6 h of treatment with dGMPSe used at concentration of 100 µM, which well corresponds to the IC_50_ value for the same parameters used in the MTT assay. When the concentration of dGMPSe was increased to 200 μM, all cells died, whereas dGMP used in a similar concentration had no effect on the viability of HeLa cells, and they were morphologically intact (phase contrast panel, Figure 7).

By comparing the phase contrast and fluorescence images, we can conclude that dead cells due to the intracellular toxic activity of selenium containing compounds (dGMPSe/H_2_Se) show higher levels of fluorescence intensity in the presence of the SF7 probe (Figure 7).

#### 2.3.2. The Accumulation of H_2_Se after Administration of dGMPSe to HeLa Cells Detected by Fluorescence

The MTT viability assay and microscopic analysis confirmed our hypothesis that 5′-*O*-selenophosphate nucleosides can be toxic to cancer cells. In the next step, we tempted to assess whether the cytotoxicity of dGMPSe was due to the release of H_2_Se.

To check this hypothesis, HeLa cells were exposed to dGMPSe in concentrations close to IC_50_ for 6 and 24 h, that is, at concentrations of 100 and 10 μM. Untreated or dGMP treated cells were used as controls. To detect the release of H_2_Se, the SF7 fluorogenic probe was added to the medium with the HeLa cells after the addition of dGMPSe and one hour before the data collection. As we have shown, SF7 can sense rapidly the H_2_S or H_2_Se presence, and therefore has been successfully used for detection both in vitro (previous section) and in cells [12]. In Figure 8, it was shown that HeLa cells treated with dGMPSe exhibited a time- and concentration-dependent increase in fluorescence. At 100 µM concentration of dGMPSe, an increase in fluorescence intensity of almost 28% (6 h) and 30% (24 h) was observed compared to the controls, and the changes were statistical significant.

However, the level of H_2_Se increased with time at 100 μM but not at 10 μM. We assume that transport without carrier is fast at the beginning (6 h), but slows down over time (24 h) as the concentration of the compound decreases. The 10 µM concentration is high enough to detect fluorescence after 6 h, but after 24 h the concentration of dGMPSe is too low for effective transport and eventually an increased amount of H_2_Se is not detected. In contrast, the initial concentration of 100 µM is sufficient for effective transport and detection after 24 h.

Observation of HeLa cells after addition of dGMPSe under invert fluorescent microscope (Figure 7) informed us that dead cells exhibited higher fluorescence intensity than living cells. Taking together, these results suggest that high level of H_2_Se led to cell death.

Passive transport of negatively charged nucleotides and oligo(nucleotide phosphorothioate)s (PS) and phosphoroselenoates (PSe) is difficult. However, there are reports demonstrating efficient transport of such “naked” PS-oligonucleotides, presumably related to their strong interaction with membrane proteins leading to their efficient cellular uptake compared to unmodified oligos [26]. The PS-oligos transported without a carrier also showed intracellular functionality [27]. Therefore, we hypothesized that monomeric selenium-containing nucleotides might have similar properties, and are able to cross cellular membrane without a carrier.

Nucleotides and nucleosides can also be introduced into cells by activation of P2X7 receptors. For example, adenosine and guanosine 5′-*O*-phosphorothioates have been used as H_2_S donors in isolated kidney glomeruli in the presence of a P2X7 receptor agonist BzATP [28]. Strong and prolonged activation of P2X7 receptors (by agonists) makes them permeable to molecules with molecular weight. up to 900 Da. Therefore, activation of P2X7 receptors may be an alternative for transport using carrier and uptake without carrier under in vitro conditions, but it is difficult to apply in vivo.

In addition, there is also the possibility of transporting dGMPS and dGMPSe through specific nucleoside/nucleotide receptors. A role for adenosine receptors (ARs) in mediating guanosine effects has been reported [29]. Therefore, perhaps other adenosine or adenosine phosphate receptors can be utilized by guanosine derivatives. Recently, Laurent and co-workers have shown that PS-oligonucleotides enter cells through thiol-mediated uptake [30]. Selenium and sulfur share many physicochemical properties, and cysteine- and selenocysteine-containing peptides can form S-S, S-Se and Se-Se bonds [31]. Therefore, it is possible that dGMPSe may be transported by a similar mechanism (mixed S-Se-mediated uptake). Consequently, transport of nucleosides phosphorothioate (and selenophosphate) without a carrier may not be difficult, and is effective enough for the compounds to be functional.

#### 2.3.3. Fluorescent Detection of H_2_Se in HeLa Cells Electroporated with dGMPSe

In the next step, we used electroporation to deliver dGMPSe directly in HeLa cells, and fluorescence intensity was measured using SF7 or SF4 probes. Untreated or dGMP-treated cells served as negative controls and GMPS was used as a positive control. Similar to the experiments with the recombinant HINT1 enzyme (Section Enzymatic assays), both probes were more sensitive to H_2_Se than to H_2_S, as shown by the comparison of fluorescence intensity for 1.5 mM GMPS and 10 µM dGMPSe. At these concentrations, both compounds showed similar values of fluorescence (Figure 9a). After 24 h, when dGMPSe was used at a concentration close to the IC_50_ (10 µM), a 50% higher fluorescence intensity was observed for the SF7 probe and a 24% higher for the SF4 probe compared to the controls (Figure 9). After 6 h, the fluorescence intensity corresponding to H_2_Se release was lower than after 24 h, and remained about 22% (SF7) and 12% (SF4) higher than for the controls, respectively. These results indicate a time-dependent activity of dGMPSe in the cells, similar to the passive transport of dGMPSe shown in Figure 8, but observed at much higher concentrations of the compound studied. Two different concentrations of dGMPSe (10 and 100 µM) were used in the experiment with the SF4 probe, and the different fluorescence values were also observed at the 6 and 24 h time points for 100 µM (12% versus 28% compared to the control). We did not observe any significant differences in fluorescence values at 24 h between the two concentrations used. This can be explained by the fact that after prolonged exposure to a concentration of 100 µM, these cells were mostly dead and their numbers were lower than those of cells treated at the same time with 10 µM, for which IC_50_ was 8 μM at 24 h (Table 3). These results indicate that H_2_Se was produced before cell death.

## 3. Discussion

Using biochemical assays (HPLC- and fluorescence-based), we investigated the enzymatic activity of HINT1 towards 2′-deoxyguanosine-5′-*O*-selenophoshate compared to the corresponding nucleoside thiophosphate, i.e., dGMPS. As we suspected, HINT1 exerted relatively similar hydrolytic activity toward dGMPSe and dGMPS due to the very similar chemical properties of selenium and sulfur. Similar values for k_cat_ (0.02 s^−1^) were found for both substrates, and only a small difference was observed for K_m_ (Table 2). During the reactions, the release of the reducing agents H_2_Se and H_2_S, respectively, was expected and detected (Figure 4).

More importantly, we also observed the release of H_2_Se or H_2_S in the cells after the administration of dGMPSe and/or dGMPS, both without a carrier (as a result transport without carrier) and by electroporation (directly into the cells) (Figure 8 and Figure 9). The amount of H_2_Se detected depended on the time, concentration and method of introduction of the compounds studied. Previously, we showed that HINT1 is involved in the desulfuration process in HeLa cells, and this dependence was demonstrated using RNA interference [12]. The cells in which HINT1 was silenced were unable to effectively desulfurate AMPS and GMPS. Therefore, our comparative experiments with dGMPSe and dGMPS suggest that the HINT1 enzyme is also capable of converting (d)NMPSe to (d)NMP and H_2_Se both in vitro and intracellularly.

According to the findings of Koziolkiewicz and co-workers [32], the influence of extracellular nucleotides and their phosphorothioate analogs depended on the activity of the ecto-5′-nucleotidase present at the cell surface. However, this activity was observed for compounds at a concentration of 200 µM after 72 h incubation with HeLa cells (inhibition of cell growth: about 15% for dGMP, 1.3% for dGMPS and 30% for dGuanosine). The cytotoxic effect of dGMPSe was observed after 12 h and at an IC_50_ concentration of 19 µM. Therefore, a possible involvement of ecto-5′-nucleotidase can be rather excluded, because the action of this enzyme, would produce dGuanosine and selenophosphoric acid (H_3_SePO_3_). As shown above, dG is not very toxic but we could not find exact data on the toxicity of selenophosphoric acid. In contrast, there is a lot of data on the toxicity of H_2_Se. Moreover, of these three compounds, only H_2_Se is the substrate for the fluorogenic SF probes, so fluorescence is only produced in cells in its presence.

On the other hand, pyrimidine nucleoside 5′-*O*-monophosphorothioates (acting as agonists of the P2Y6 receptor) have been reported to stimulate P2Y6 activation of cell migration in HeLa cells [33]. We have not performed any studies to observe the migration of Hela cells after the application of dGMPSe, especially due to the high toxicity of this compound; perhaps this could be a subject for further studies.

Selenium and sulfur, while similar, also exhibit significant differences. H_2_S and H_2_Se both exhibit reducing properties, but H_2_S has a longer lifetime in cells than H_2_Se, which is due to the higher reactivity of the latter. Nucleoside thiophosphates are not toxic to cells (or show very low cytotoxicity): HeLa cells treated with dGMPS at a concentration of 200 µM were still more than 80% viable after 48 h (data not shown). In contrast, dGMPSe at a concentration of 19 µM induced the death of half of the HeLa cells after only 12 h (Table 3, Figure 6a); this implies that dGMPSe or the product of its enzymatic hydrolysis H_2_Se is highly toxic to cancer cells. Thus, the MTT viability assay and microscopic analysis confirmed our hypothesis that selenophosphate nucleosides can be toxic to cancer cells, and our further tests showed that this toxicity was due to the release of H_2_Se.

Previous studies have shown that the cytotoxicity and anticancer effects of selenium compounds are mainly due to their prooxidant actions, including the oxidation of protein thiols and the formation of ROS by the rapid reaction of H_2_Se with O_2_ and the formation of elemental selenium and superoxide anion radicals (O_2_^▪−^) [6,7,34]. The generated ROS species cause a decrease in mitochondrial membrane potential, leading to the release of cytochrome c into the cytosol, which in turn results in cell apoptosis. It has also been shown that pharmacological concentrations of Na_2_SeO_3_ do not induce oxidative stress under hypoxic conditions, but instead induce reductive stress through the formation of H_2_Se as a result of Na_2_SeO_3_ metabolism and low O_2_ [8]. While an excess of ROS triggers oxidative stress, their continued depletion leads to the opposite condition, reductive stress. If left unaddressed, reductive stress can block cell differentiation and lead to cell death by inducing autophagy or apoptosis [8]. The reductive stress response relies on the increase of mitochondrial membrane potential, the restoration of mitochondrial activity in cells by turning on/accelerating oxidative phosphorylation, which yields ROS as invariant byproducts. This activity can restore redox homeostasis in cells. Therefore, a low level of reductive compounds can be compensated by cells, but their high level can lead to redox imbalance and cell death. The question is whether the presence of a high content of compounds with strong reductive properties can intercept ROS and cause their deficiency under extreme/oxygen conditions, which can also lead to reductive stress.

The activity of various selenium compounds depends on the formation of hydrogen selenide, the actual trigger of cell death. Many selenium compounds, including those from food, are metabolized to H_2_Se by the GSH reduction system and enzymes of cysteine metabolism [4]. Here we introduce new selenium compounds, nucleoside 5′-*O*-selenophosphates, which require a different metabolic pathway involving the enzyme HINT1 to be converted to H_2_Se. Thus, selenium-containing nucleotides are another option for compounds with anticancer therapeutic potential.

## 4. Materials and Methods

### 4.1. Chemistry

#### 4.1.1. General Methods

NMR measurements were performed using a Bruker Avance III 500 spectrometer (Bruker Corporation, Billerica, MA, USA), operating at 500.1, 125.8 and 202.45 MHz for ^1^H, ^13^C and ^31^P, respectively.

Mass spectra (HRMS) were recorded using an Agilent 6520 Q-TOF LCMS (Agilent Technologies, Santa Clara, CA, USA). Analytical RP-HPLC (Waters, Corp., Milford, MA, USA) was used to determine the purity of compound **4** [Grace LC-18 column (4.6 × 250 mm), flow rate 1 mL/min, buffer A, 0.05 M TEAB pH 7.5; buffer B, 40% CH_3_CN in 0.05 M TEAB; gradient from 0 to 40% B over 30 min].

#### 4.1.2. Synthesis of dGMPSe

5′-*O*-Selenophosphate-2′-deoxyguanosine sodium salt (**4**): 3′-*O*-Acyl-*N*2-iBu protected 2′-deoxyguanosine **1** (280 mg, 0.740 mmol (Figure 1), anhydrous CH_2_Cl_2_ (7 mL), DIPEA (258 µL, 1.48 mmol) and elemental selenium (58 mg, 0.74 mmol) were placed in a round bottom flask (25 mL) and 2-chloro-4,4-pentamethylene-1,3,2-oxathiaphospholane **2** (184 mg, 0.870 mmol) was added dropwise with stirring. The reaction mixture was stirred for 12 h at room temperature. The solvent was removed in vacuo and the residue was dissolved in chloroform (2–3 mL), and subjected to a silica gel column chromatography (2.5 × 18 cm), methanol in chloroform (0→10 %) to afford **3** (220 mg, 0.348 mmol, 47%) as a white foam. A round bottom flask (25 mL) was charged with **3** (220 mg, 0.348 mmol) and anhydrous 3-hydroxypropionitrile (48 µL, 0.70 mmol). DBU (106 μL, 0.700 mmol) was added, and the reaction mixture was stirred at room temperature for 12 h. The solvent was removed by rotary evaporation, and the residue was dissolved in potassium carbonate in methanol (0.05 M, 5 mL, removal of the protecting groups). The solvent was removed by rotary evaporation. The purified compound was obtained by Sephadex A-25 ion-exchange chromatography using a linear gradient of ammonium bicarbonate buffer (pH 7.5) as eluent to give **4** as the nucleoside ammonium salt, which was converted to its sodium salt (Dowex 50 WX4 Na^+^): (43.0 mg, 0.104 mmol, 30%). White foam. HRMS (ESI-TOF) [M-H]^−^ calculated for C_10_H_13_N_5_O_6_PSe 409.9778, found 409.9774. ^1^H NMR (D_2_O, δ): 8.12 (s, 1H, H-8), 6.21 (t, *J* = 6.5 Hz, 1H, H-1′), 4.70 (d, *J* = 2.2 Hz, 1H, H-3′), 4.23 (br s, 1H, H-4′), 4.05–3.98 (m, 2H, H-5′, H-5”), 2.78–2.70 (m, 1H, H-2”), 2.51–2.45 (m, 1H, H-2′); ^13^C 158.71, 153.67, 151.05, 137.56, 115.87, 85.85, 83.46, 71.66, 64.52, 58.92; ^31^P 29.80. RP-HPLC: Rt = 10.61 min.

### 4.2. Expression and Purification of HINT1 and HINT2

Human HINT1 protein was expressed in bacteria *E. coli* BL21* using the pSGA02-hHINT1 plasmid and purified by AMP -agarose (matrix spacer 8 over the C-8 atom of 5′-AMP, SIGMA, Oakville, ON, USA) affinity chromatography as previously described for rabbit Hint1 [15]. Human HINT2 was expressed in *E.coli* using the pGAT2-HINT2 vector and purified in two steps as described [21]. Briefly, the His6-GST-tagged HINT2 obtained in the first step was purified by Ni- IDA agarose chromatography (Qiagen, Germantown, MD, USA), and then after cleavage by thrombin (to remove the His6-GST-tag) by AMP-agarose chromatography. Lysis and purification of both proteins was performed with the addition of protease inhibitor cocktail (Roche Diagnostics, Mannheim, Germany). Finally, the homogeneous enzyme preparations were concentrated and stored at −80 °C. The purity of the proteins was evaluated by SDS-PAGE analysis and subsequent staining with the PAGE Blue Protein Staining Solution (Fermentas, Lithuania).

### 4.3. HPLC Enzymatic Assay

Substrates at a concentration of 50 μM were incubated with HINT1 or HINT2 in 20 μL of buffer containing 20 mM HEPES pH 7.2 and 0.5 mM MgCl_2_ at 37 °C for 5–15 min. The reaction mixtures were analyzed by RP-HPLC using a Kinetex C18 column (2.6 µ C18, 100 A, 100 × 2.10 mm, Phenomenex, Torrance, CA, USA) eluted with buffers (A) 50 mM TEAB pH 7.5 and (B) 40% CH_3_CN in 50 mM TEAB, isocratic 9% B for 8 min then on a gradient of 9–100% B for 8 min and 100% B for 2 min, at a flow rate of 1 mL min^−1^. Quantification was performed by integrating the peak areas of the substrates and the products at spectrophotometrically at 254 nm.

To determine the kinetic parameters for the hydrolysis of dGMPSe and dGMPS, the initial rate assays were performed at 37 °C in volumes of 10–200 μL of the buffer indicated above at the various substrate concentrations 25, 50, 100, 200, 300, 400, 500, 800, 1000 μM with the addition of 100–300 pmol of the enzyme. The products were analyzed using RP-HPLC as previously described. Under these conditions, the retention times (Rt) of the substrate (dGMPSe), reference compound (dGMPS) and product (GMP) were 3.864 min, 4.766 min and 4.729 min, respectively (Figure 2). To calculate K_m_ and k_cat_ values, the concentrations (X) and corresponding observed reaction rates (Y) were processed using a nonlinear Michaelis-Menten regression Y = Vmax × X/(K_m_ + X). Reactions were performed in triplicate and data were processed using GraphPad Prism 5.0 software (La Jolla, CA, USA).

### 4.4. Fluorescent Enzymatic Assay

Substrates at a concentration of 200 μM were mixed with HINT1 (73 pmol) or HINT2 (146 pmol) in 50 μL of buffer containing 20 mM HEPES pH 7.2 and 0.5 mM MgCl_2_ with addition of SF7 (Sulfidefluor 7 AM; Tocris Bio-Techne, Bristol, UK) or SF4 (Sulfidefluor 4, synthesis according to ref. [23]) at concentration of 5 µM. Reaction mixtures were prepared in triplicate in black 96-well plate (half-area, flat bottom; Costar, Washington, DC, USA) and placed in a Synergy HT plate reader with temperature control (BIO-TEK, Winooski, VT, USA) equipped with 485/528 ± 20 nm filters (ex/em). The reactions were performed at a temperature of 37 °C. Fluorescence values were measured at the indicated time points (every 5 or 15 min). The amount of fluorescence, relative to the content of H_2_S (for GMPS and dGMPS substrates) or H_2_Se (for dGMPSe substrate), was evaluated as relative light units. Reactions without any substrate (but with the enzyme in buffer), with dGMP (an additional negative control) and with dGMPSe in buffer were only used as controls.

### 4.5. Cell Lines and MTT Assay

HeLa cells (human cervical carcinoma, ECACC 93021013) were cultured in RPMI 1640 medium (Gibco, Life Technologies, Grand Island, NY, USA) supplemented with 10% FBS (Gibco, Life Technologies, Grand Island, NY, USA) and antibiotics (100 U/penicillin and 100 mg/mL streptomycin; Gibco, Life Technologies, Grand Island, NY, USA). Cells were maintained in a humidified atmosphere of 5% CO_2_, 95% air at 37 °C.

For the MTT assay, cells were seeded to a 96-well plate in the exponential growth phase at a concentration of 10 × 10^3^ per 200 µL in complete medium and allowed to adhere for 16–24 h. The tested compounds dGMPSe and dGMP were added at the final concentrations of 0.1, 1, 10, 50 and 100 µM and the number of cells was determined after 12 h, 24 h and 48 h using the tetrazolium salt method (MTT assay) [24] with following modification. After the indicated time, the medium was replaced with a fresh one before adding the MTT substrate (3-(4,5-dimethylthiazol-2-yl)-2,5-diphenyltetrazolium bromide). Each MTT test point represents the mean ± SE of at least two independent measurements made in quadruplicate.

### 4.6. H_2_Se and H_2_S Cellular Assay

Twenty-four hours before transfection, cells were transferred to a new vial so that they were in logarithmic growth on the day of transfection. HeLa cells were transfected with dGMPSe or GMPS (positive control) by electroporation (Nucleofector II, Lonza, Slough, UK) using PBS as electroporation buffer and electroporation parameters specific for HeLa cells line according to the manufacturer’s protocol. On the day of transfection, cells were harvested with trypsin 0.05% and counted in Burker’s chamber, and 0.5 × 10^6^ cells were used for an electroporation experiment (for one compound) with 100 μL volume in a sterile disposable cuvette. After transfection, 500 μL of prewarmed medium was added, and the cells were transferred to a 6-well plate containing 1.5 mL of culture medium with the addition of a fluorogenic probe (5 μM SF4 or 2.5 μM SF7) per well. Cells from one well (total volume 1.6 mL) were divided, added to a 96-well plate (~125,000 cells at 200 μL per well), and cultured. Fluorescence levels were measured every two hours using a Synergy HT plate reader (BIO-TEK, Winooski, VT, USA) equipped with 485/528 ± 20 nm filters (ex/em). The amount of fluorescence corresponding to the content of H_2_Se or H_2_S, was assessed as relative light units. Cells transfected with H_2_O only were used as controls. As an additional negative control, dGMP was used, while GMPS served as a positive control.

### 4.7. Invert Microscope Imaging of Living Cells

Fluorescence of living HeLa cells was analyzed by fluorescence microscopy. Cells were grown on 96-well plates with clear flat bottom and black walls. At 5 h after treatment with dGMPSe, 2.5 μM SF7 probe or 5 µM SF4 was added to the medium, and the cells were incubated at 37 °C for 1 h. Then, cells were imaged using a fluorescence microscope (Nikon Eclipse Ti-U inverted microscope; Nikon Europe BV, Amsterdam, The Netherlands) with a 465/495 nm excitation filter and detected with a 515/555 nm emission filter. Alternatively, cells were washed with PBS buffer and then imaged immediately. Images were taken for three selected fields per well (magnification × 20) and compared at similar exposure times (snapshot). Analysis of the images was performed using NIS element BR 3.0 basic software (Nikon Europe BV, Amsterdam, The Netherlands).

### 4.8. Statistical Analysis

For statistical analysis of data from fluorescence measurement experiments, the fluorescence of the background (a medium containing the probe SF) was subtracted and the data were calculated using the following equation: ((fluorescence of probe-fluorescence of control)/fluorescence of control) × 100%. Results were expressed as % fluorescence with respect to H_2_S or H_2_Se content compared to the control. Data are the means ± SEM of two or three independent experiments performed in five or more replicates. The significance of the data was determined by a t-test using GraphPad Prism 5.0 software (La Jolla, CA, USA). A *p*-value of ≤0.05 for comparisons between groups was considered statistically significant.

## Data Availability

Not applicable.

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
