# Peer review of "Intracellular HINT1-Assisted Hydrolysis of Nucleoside 5′-O-Selenophosphate Leads to the Release of Hydrogen Selenide That Exhibits Toxic Effects in Human Cervical Cancer Cells"

_ijms, 2022, doi:10.3390/ijms23020607_

Round 1

Reviewer 1 Report

The authors describe their results on the synthesis and biochemical assays of a new selenium derivative, 2'-deoxyguanosine-5'-O-selenophosphate (dGMPSe). By the application of biochemical assays (HPLC- and fluorescence-based ones) they were able to investigate the ability of the histidine triad superfamily enzymes HINT1 and 2 towards the above compound and its sulfur analogue. HINT1 effectively hydrolyzes both compounds either in passive transport or directly in the cells and generate hydrogen selenide and sulfide, respectively. The former was found to be an effective anticancer agent against HeLa cells in the low micromolar range. The manuscript constitutes novel pieces of information on a selenium-containing nucleotide with anticancer therapeutic potential.

The manuscript is recommended for publication after rectifying a few minor issues listed hereunder:

P. 3.,line 97: "PIII" should be replaced by "P(III)".
P. 4, line 139: "one order of magnitude" should be replaced "by one order of magnitude".
P. 5, Table 1 heading: the hydrolysis rate unit should be properly written as in line 137.
P. 5, lines 177-181: reversible arrows should be used appropriately. If you are using MS Word, you may consult these sources: https://www.youtube.com/watch?v=IXML84P3P-c; https://libanswers.walsh.edu/faq/147998; https://support.office.com/en-us/article/write-an-equation-or-formula-1d01cabc-ceb1-458d-bc70-7f9737722702.
P. 10 and 11, Figures 8 and 9: it is an unfortunate choice to use two blue and green colours with similar shade.
P. 15, ref. 9: delete "pii".
P. 16, ref. 21: use proper abbreviation "Biochim. Biophys. Acta, Gen. Subj." (cf. https://cassi.cas.org/publication.jsp?P=eCQtRPJo9AQyz133K_ll3zLPXfcr-WXfYemHjqucFn-d7P7RDj_heAPfSAVi1ul7SqJWEZ7F36kyz133K_ll3zLPXfcr-WXfXT3to5uCWtUyz133K_ll3zLPXfcr-WXfFV5Qn1sQu8BsDpuJibhy6A)
P. 16, ref. 24: use inclusive pagination "203-210".

Author Response

Response to the reviewers' suggestions

Reviewer #1: Thank you very much for the review. We have changed our manuscript according to the reviewer's suggestions and all changes are marked in blue.

3.,line 97: "PIII" should be replaced by "P(III)" - DONE
P. 4, line 139: "one order of magnitude" should be replaced "by one order of magnitude". - DONE
P. 5, Table 1 heading: the hydrolysis rate unit should be properly written as in line 137. - DONE
P. 5, lines 177-181: reversible arrows should be used appropriately. - DONE
P. 10 and 11, Figures 8 and 9: it is an unfortunate choice to use two blue and green colours with similar shade. - We have  changed the pattern of the bars with similar colours in Figures 8 and 9 to better distinguish these groups.

P. 15, ref. 9: delete "pii". - DONE
P. 16, ref. 21: use proper abbreviation "Biochim. Biophys. Acta, Gen. Subj." - DONE
P. 16, ref. 24: use inclusive pagination "203-210". - DONE

Reviewer 2 Report

The article "Intracellular HINT1-assisted hydrolysis of nucleoside 5’-O-selenophosphate leads to the release of hydrogen selenide that exhibits toxic effects in human cervical cancer cells” attempts to provide insight into cytotoxic properties of nucleoside 5’-O-selenophosphates. The authors demonstrate the release of toxic H2Se from dGMPSe both in vitro and inside HeLa cells.

The text is well organized. The manuscript has a good experimental design and reporting. However, I have several concerns that should be explained:

1. The authors state that dGMPSe can be transported into HeLa cells by the P2X7 receptor and cite the paper by Beltowski et al. to support this hypothesis. In the cited article, it was shown that treatment of isolated glomeruli with AMPS or GMPS resulted in no detectable H2S production. However, H2S accumulation was detected if glomeruli were incubated in the presence of nucleotides and P2X7 receptor agonist, BzATP. H2S production by glomeruli incubated with AMPS or GMPS and BzATP was completely abolished by P2X7 receptor antagonist, A-438079. Indeed, strong and prolonged P2X7 receptor activation makes them permeable to molecules up to 900 Da molecular weight.

Therefore, how do authors explain the transport of dGMPSe without activation of P2X7?

2. I would like to know whether the authors consider other mechanisms of action of nucleoside 5’-O-selenophosphates. Assuming that transport is difficult, part of dGMPSe may remain in the extracellular environment. In the case of phosphorothioate nucleotides, the involvement of nucleotide receptors has been proven. Moreover, in the case of HeLa cells, such P2Y activation stimulated cell migration. Do the authors consider such possibility also in the case of nucleoside 5’-O-selenophoshosphates. Please discuss this issue.

3. Why GMPS (not dGMPS) was used as a positive control for dGMPSe in fluorescent detection of H2Se in HeLa cells electroporated with dGMPSe

4. Hint1 and HINT1 are used within the text. Why?

5. Add hydrolysis rate of AMPS to Table 1

6. Line 248

The viability of HeLa cells was determined at five different concentrations of dGMPSe: 0.1 μM, 1 μM, 10 μM, 50 μM and 100 μM.

Figure 6 indicates different concentrations

7. Figure 8 place **** just above bars

Apply the same y values (0-45)

The level of H2Se increased with time for 100 μM and not for 10 μM. Do the authors have the hypothesis why?

Author Response

Reviewer #2

Thank you very much for the review. All the introduced changes in the main text have been marked in red.

  1. The authors state that dGMPSe can be transported into HeLa cells by the P2X7 receptor and cite the paper by Beltowski et al. to support this hypothesis. In the cited article, it was shown that treatment of isolated glomeruli with AMPS or GMPS resulted in no detectable H2S production. However, H2S accumulation was detected if glomeruli were incubated in the presence of nucleotides and P2X7 receptor agonist, BzATP. H2S production by glomeruli incubated with AMPS or GMPS and BzATP was completely abolished by P2X7 receptor antagonist, A-438079. Indeed, strong and prolonged P2X7 receptor activation makes them permeable to molecules up to 900 Da molecular weight.

Therefore, how do authors explain the transport of dGMPSe without activation of P2X7?

  • Perhaps we have really gone too far with the possibility of transporting our compounds via the P2X7 receptor, because this receptor should be activated (by ATP or a receptor agonist such as BzATP). And "strong and prolonged P2X7 receptor activation makes them permeable to molecules up to 900 Da molecular weight" - this situation does not occur in our case, this is just a possibility. Therefore, we have removed this part from our manuscript.

However there is still a part about the transport of “naked” PS-oligonucleotides and phosphorothioate nucleotides:

“Passive transport of negatively charged nucleotides and oligo(nucleotide phosphorothioate)s (PS) and phosphoroselenoates (PSe) is difficult. However, there are reports demonstrating efficient transport of such “naked” PS-oligonucleotides, presumably related to their strong interaction with membrane proteins leading to their efficient cellular uptake compared to unmodified oligos [J. Am. Chem. Soc., 2020, 142, 7456-7468]. The PS-oligos transported without a carrier also showed intracellular functionality [Biochem. Biophys. Res. Commun., 2021, 573, 140-144]. …Therefore, we hypothesized that monomeric selenium-containing nucleotides might have similar properties and are able to cross cellular membrane without a carrier.”(page 10, lines 305-314).

In addition, there is also the possibility of transporting dGMPS and dGMPSe through specific nucleoside/nucleotide receptors. A role for adenosine receptors (ARs) in mediating guanosine effects has been reported [Lanznaster D. et al. Cells 2019, 8, 1630]. Therefore, perhaps other adenosine or adenosine phosphate receptors can be utilized by guanosine derivatives. Recently, Laurent and co-workers have shown that PS-oligonucleotides enter cells through thiol-mediated uptake [Angew Chem Int Ed Engl. 2021; 60(35):19102-19106]. Selenium and sulfur share many physicochemical properties, and cysteine- and selenocysteine-containing peptides can form S-S, S-Se, and Se-Se bonds [ J. Am. Soc. Mass Spectrom. (2012) 23, 11, 2001-2010]. Therefore, it is possible that dGMPSe may be transported by a similar mechanism (mixed S-Se-mediated uptake). Consequently, transport of nucleosides phosphorothioate (and selenophosphate) without a carrier may not be difficult and is effective enough for the compounds to be functional.

This explanation has been added to the main text.

  1. I would like to know whether the authors consider other mechanisms of action of nucleoside 5’-O-selenophosphates. Assuming that transport is difficult, part of dGMPSe may remain in the extracellular environment. In the case of phosphorothioate nucleotides, the involvement of nucleotide receptors has been proven. Moreover, in the case of HeLa cells, such P2Y activation stimulated cell migration. Do the authors consider such possibility also in the case of nucleoside 5’-O-selenophoshosphates. Please discuss this issue.

In our article, we showed that some properties of dGMPS and dGMPSe are similar. These probably include interaction with nucleotide receptors and "In the case of phosphorothioate nucleotides, the involvement of nucleotide receptors has been proven", so yes, based on this similarity, we think a similar transport mechanism of both compounds is very likely and we pointed it out in our article (page 10, line 313).

 We have added an appropriate explanation in the main text (see above).

According to the findings of Koziolkiewicz and co-workers [Blood, 2001, 98:995–1002], the influence of extracellular nucleotides and their phosphorothioate analogs depended on the activity of the ecto-5'-nucleotidase present at the cell surface. However, this activity was observed for compounds at a concentration of 200 µM after 72 h incubation with HeLa cells (inhibition of cell growth: about 15% for dGMP, 1.3% for dGMPS and 30% for dGuanosine). The cytotoxic effect of dGMPSe was observed after 12 hours and at an IC50 concentration of 19 µM. Therefore, a possible involvement of ecto-5'-nucleotidase can be rather excluded, because the action of this enzyme, would produce dGuanosine and selenophosphoric acid.  As shown above, dG is not very toxic but we could not find exact data on the toxicity of selenophosphoric acid H3SePO3. In contrast, there is a lot of data on the  toxicity of H2Se. Moreover, of these three compounds, only H2Se is the substrate for the fluorogenic SF probes, so fluorescence is produced in cells only in its presence.

We have added an appropriate explanation in the main text (Discussion section).

On the other hand, pyrimidine nucleoside 5'-O-monophosphorothioates (acting as agonists of the P2Y6 receptor) have been reported to stimulate P2Y6 activation of cell migration in HeLa cells [Purinergic Signalling (2016) 12:199–209]. We have not performed any studies to observe the migration of Hela cells after the application of dGMPSe, especially due to the high toxicity of this compound. Perhaps, this could be a subject for the further study.

  1. Why GMPS (not dGMPS) was used as a positive control for dGMPSe in fluorescent detection of H2Se in HeLa cells electroporated with dGMPSe

GMPS is a good positive control because it gives a relatively high fluorescence signal (after 6 hours: about 35% at a concentration of 1.5mM with the SF7 probe and about 10% at a concentration of 100µM with the SF4 probe). On the other hand, the fluorescence values after 24 hours and dGMPS at a concentration of 1mM using the SF7 probe were about 10% [Biochem Pharmacol., 2019, 163, 250-259]. It can be concluded that dGMPS is not a good positive control as the results obtained are close to the limit of quantification. Due to better reactivity of H2Se than H2S towards SF probes, the results for dGMPSe are statistically significant.

  1. Hint1 and HINT1 are used within the text. Why?

According to widely accepted nomenclature, HINT1 means human HINT1 and Hint1 has a broader meaning (it could also be the enzyme of other species, e.g. from rabbit).

We have added an appropriate explanation in the main text.

  1. Add hydrolysis rate of AMPS to Table 1

We have added this value.

  1. Line 248: The viability of HeLa cells was determined at five different concentrations of dGMPSe: 0.1 μM, 1 μM, 10 μM, 50 μM and 100 μM.

Figure 6 indicates different concentrations.

We have made these concentrations compatible with Figure 6 (the 200 µM values have been removed because these data were only available after 24 hours and there was no difference in cell viability between 100 and 200 µM dGMPSe at that time).

  1. Figure 8 place **** just above bars

We placed **** above the bars as it was possible using our software.

  1. Apply the same y values (0-45)

We have done it.

  1. The level of H2Se increased with time for 100 μM and not for 10 μM. Do the authors have the hypothesis why?

I assume that this question refers to Figure 8 (transport without carrier), because in Figure 9 (after electroporation) the H2Se content increases with time at both concentrations.

We assume that transport without carrier is fast at the beginning (6h), but slows down over time (e.g. 24h) as the concentration of the compound decreases. The 10 µM concentration is high enough to detect fluorescence after 6h, but after 24h the concentration of dGMPSe is too low for effective transport and eventually an increased amount of H2Se is not detected. In contrast, the initial concentration of 100 µM is sufficient for effective transport and detection after 24h.

We have added an appropriate explanation in the main text (Results, page 9).

Round 2

Reviewer 2 Report

The authors improved the manuscript but

  1. They have not added an appropriate explanation in the main text (Discussion section) as stated in the response section.

On the other hand, pyrimidine nucleoside 5'-O-monophosphorothioates (acting as agonists of the P2Y6 receptor) have been reported to stimulate P2Y6 activation of cell migration in HeLa cells [Purinergic Signalling (2016) 12:199–209]. We have not performed any studies to observe the migration of Hela cells after the application of dGMPSe, especially due to the high toxicity of this compound. Perhaps, this could be a subject for the further study.

2. I suggest not removing the part describing the P2X7 receptor. Instead, authors could discuss the limitation of this hypothesis.

Author Response

 Response to the Reviewer #2

Thank you very much for the review. We have changed our manuscript according to the reviewer's suggestions and all changes are marked in red.

Ad.1. We have added this part to the main text (Discussion section, page 12).

Ad.2. We have made as reviewer suggested. This part has been placed in Results section, page 10.